# Does Pandemic Fatigue Prevent Farmers’ Participation in the Rural Tourism Industry: A Comparative Study between Two Chinese Villages

**DOI:** 10.3390/ijerph20010062

**Published:** 2022-12-21

**Authors:** Mengyuan Qiu, Yueli Ni, Sulistyo Utomo

**Affiliations:** 1College of Economics and Management, Nanjing Forestry University, Nanjing 210037, China; 2Nanjing Institute of Tourism and Hospitality, Nanjing 211100, China; 3Griffith Business School, Griffith University, Gold Coast Campus, Gold Coast, QLD 4222, Australia

**Keywords:** physical fatigue, mental fatigue, rural tourism participation, Chinese villages, COVID-19 pandemic

## Abstract

Rural tourism is an important income generation method for farmers post-pandemic. However, few studies have focused on how pandemic fatigue has affected their willingness to participate in rural tourism development. We conducted a quasi-experiment to test these effects using data from two Chinese villages. Shanlian village, which was more severely affected by COVID-19, was the experimental group, while Huashu village was set as the control group. Our results reveal that both physical and mental fatigue hinder farmers’ intention to engage in rural tourism. Further, there were significant interaction effects between physical and mental fatigue on the farmers’ participation in rural tourism. For farmers with low physical fatigue, the higher their mental fatigue, the less willing they were to participate in rural development. Conversely, for the higher physical fatigue group, farmers with low levels of mental fatigue were still more willing to participate in rural tourism development. These findings reduce the current research gap concerning the relationship between pandemic fatigue and farmers’ participation in rural tourism and indicate that practitioners and policymakers should consider farmers’ fatigue management as an important factor for the sustainability of rural tourism during the ongoing COVID-19 crisis.

## 1. Introduction

Pandemic fatigue is an emerging health concern among individuals during the COVID-19 crisis [1,2]. Evidence shows approximately 63% of individuals in the post-COVID-19 period experienced physical tiredness, body weakness, sleep problems, mental distress, loneliness, boredom, and extreme sadness after the initial implementation of the national lockdown and mobility restrictions [3]. Accounting for 80% of China’s population, farmers face a tremendous amount of epidemic threat due to several factors, including behindhand economic and natural limits, poor health and medical conditions, a lack of relevant experience for self-protection, and even conservative ideology and behavioral modes [4]. This was seen in many villages where farmers became increasingly fatigued as the lockdown continued, preventing them from participating enthusiastically in rural tourism for income generation [5].

Community participation is the process by which people, families, or societies take up responsibility for their well-being and build the capacity to contribute to their growth and the society’s growth [6]. It is generally regarded as the most effective way for farmers to benefit from rural tourism development [7], but does not only refer to farmers working in rural tourism-related jobs. Over recent decades, the concept of participation in rural tourism research has gradually penetrated the entire process of tourism planning and emphasized farmers’ rights in the decision-making of rural tourism [8]. Farmers’ participation in rural tourism leads to changes in their livelihoods, lifestyles, social networks, and living environments [9]. Furthermore, enlisting the support and participation of farmers in rural tourism is an essential prerequisite for the revitalization of already-declining traditional agriculture [10,11]. Collectively, farmers were considered important stakeholders in the process, and emphasis was placed on the need to seriously consider their opinions and attitudes toward rural development [12,13,14].

Rural tourism describes a range of tourism activities which take place in the countryside. The impact of COVID-19 on tourism industry is going to be uneven in space and time [15]. On a detailed look, it can be observed that the damage was mainly suffered by providers in localities and regions that benefit from international tourism. On the contrary, rural tourism sites focused mainly on domestic tourism recorded only modest loss. Moreover, recent scholarly literature has even stated that the COVID-19 pandemic has had a positive effect on rural tourism due to the increase of tourists seeking for a safe/low-risk destination in scarcely populated areas [5]. However, on the supply side, the pandemic has had a negative impact on farmers who are the service providers and operation managers [16]. Since the emergence of COVID-19 in late 2019, numerous farmers have experienced a significant amount of pandemic fatigue after the implementation of a national lockdown or curfew order in China. According to the conservation of resources theory (COR), when resources are outstretched or exhausted, individuals enter a defensive mode to preserve the self, which is often a negative reaction to stressors [17]. Therefore, a high degree of pandemic fatigue occurs when central or key resources are threatened with loss, which may adversely affect farmers’ enthusiasm and willingness to engage in rural tourism. COVID-19 outbreaks occur repeatedly, with short intervals and long durations across the globe, consequently posing a new challenge for all of humanity. Two pertinent questions remain unanswered in previous literature: are farmers’ pandemic fatigue induced by the COVID-19 epidemic a predictor of their willingness to participate in rural tourism development? How does this pandemic fatigue act as a potential constraint for farmers’ support and involvement in rural tourism development?

Considering the potentially negative consequences of the outbreak, this study aimed to investigate how pandemic fatigue impacts farmers’ participation in rural tourism. A quasi-experiment on two Chinese villages which were differentially affected by the COVID-19 crisis allowed us to investigate whether comparable effects were observed in different contexts and confirm the stability and specificity of the relationship between pandemic fatigue and farmers’ participation in rural tourism [18]. Knowledge about how physical and mental fatigue prevents farmers from participating in the rural tourism industry is essential because of the increased levels of anxiety and stress that have been reported in association with the outbreak of COVID-19. The results will provide a new lens for the sustainable development of rural tourism through farmers’ fatigue management. It will gradually affect farmers’ health and well-being, as well as the economic, cultural, social, and governmental systems, and thus promoting Chinese rural revitalization.

## 2. Literature Review and Hypotheses

### 2.1. Pandemic Fatigue and Farmers’ Participation in Rural Tourism Industry

WHO describes pandemic fatigue as demotivation to engage in protection behaviors and seek COVID-19 related information due to unresolved and continuous adversity in life [19]. Correspondingly, pandemic fatigue is different from general fatigue, which may arise for various reasons and may affect people’s engagement in many activities. Notably, the introduced definition of pandemic fatigue highlights information seeking as a health-protective behavior [20]. The COVID-19 continues to affect increasingly across the world and individuals without a distinct health issue have also been affected by the pandemic. All groups can get affected by pandemic fatigue, but its severity is higher in rural areas where more aged population and chronic diseases exist. Scholars coined various terms such as “quarantine fatigue”, “behavioral fatigue”, “emergency/public/adherence fatigue”, “pandemic burnout”, and “pandemic fatigue” [21,22]. Pandemic fatigue has been well-accepted as a concept and is usually described as physical and/or mental fatigue in the existing literature [23]. Morgul investigated psychological COVID-19-related fatigue in Turkey and found that 64.1% of participants reported having physical and mental fatigue [23]. Physical symptoms of pandemic fatigue include reduction in the capacity to perform physical work resulting from activities requiring physical effort [24]. Mental fatigue caused by COVID-19 is a universal phenomenon that results from prolonged periods of task-demanding mental activity which can be subjectively described as a mental state of feeling tired or inactive [25].

Factors that impact farmers’ participation have been widely discussed in the rural tourism literature [26]. The social exchange theory (SET) explains that if farmers perceive the benefits of rural tourism development to exceed costs, they will engage in a process of exchange and support tourism development in their community [27]. For example, an increasing number of farmers are willing to participate in rural tourism since they realize that rural tourism benefits local communities by providing farming craft with a supplementary source of income and the opportunity to re-evaluate their heritage symbols and identity [28,29]. Conversely, when the costs exceed the benefits, farmers’ initial passion and eagerness gradually diminish and an overall exhaustion arises, therefore, they are reluctant to participate in rural tourism industry. During the COVID-19 crisis, the fear of restrictions such as isolation, social distancing, quarantine, and lockdown as well as uncertainty about how the pandemic would proceed may adversely affect many individuals physically and psychologically. According to the COR, this fatigue is a gradually emerging subjective state of weariness and exhaustion from COVID-19 which will prevent farmers from participating in the rural tourism industry.

For the tourism industry, a recent study confirmed that fear of COVID-19-induced shutdown negatively affects the work engagement of cruise ship employees, as much as in hotels, travel agencies, and entertainment services [30]. Another study on the tourism industry also revealed that pandemic fatigue prevents individual, occupational, and industry-level tourism participation [31]. A worker survey in the United States indicated that 25% of workers are expected to continue remote working, either part-time or full-time, due to the outbreak of COVID-19 [21]. Since rural tourism operators tend to provide face-to-face services to showcase local agricultural products and cultural activities, work exposure and working conditions increase risks to their safety, health, and wellbeing. Consequently, some farmers may be reluctant to participate in the rural tourism industry [22]. Therefore, we propose the following hypothesis:

**H1.** 
*Mental fatigue caused by COVID-19 pandemic negatively affects farmers’ participation in rural tourism development.*


**H2.** 
*Physical fatigue caused by COVID-19 pandemic negatively affects farmers’ participation in rural tourism development.*


### 2.2. The Interaction between Physical Fatigue and Mental Fatigue

Rural tourism is an industry with specific requirements for managing workers’ physical and mental fatigue status [32]. There are many types of rural tourism activities that require intensive physical and cognitive efforts. For example, as farmers participate in rural tourism development, they are often required to perform physically intensive tasks while serving tourists face-to face, and need to perform cognitive functions for tourism planning and management simultaneously. Compared with urban cities, rural areas are relatively underdeveloped in economic and social conditions. The majority of the farmers do not have accesses to a major media outlet through which the national health authority passes the appropriate preventive and control measures of COVID-19. Uncertainties about how long the COVID-19 pandemic will last, how the virus is being transmitted, how long government restrictions will last, and what will happen in the future significantly affect farmers at physical and mental levels [20]. Thus, the association and interaction between physical and mental fatigue should be considered when managing farmers who engage in rural tourism development, particularly during the COVID-19 crisis.

Though there is a lack of studies focusing on the relationship between farmers’ mental and physical fatigue during COVID-19 pandemic, studies conducted with healthcare providers, workers, young students, and elderly people demonstrated that mental and physical fatigue are related. Previous research studied the effects of various types of physical fatigue on mental workload and cognitive performance by adopting heart rate variability and a visual analogue scale [33]. Moore et al. confirmed the negative effects of exercise-induced fatigue on cognitive function through a series of tests (i.e., the visual perceptual discrimination test, memory-based vigilance test, and visual perceptual discrimination test) [34]. Mehta and Parasuraman investigated the contribution of mental fatigue to the development of voluntary physical fatigue, using a neuroergonomic approach [35]. Pageaux et al. confirmed the relationship between mental fatigue and the performance of 170 vastus lateralis muscles during cycling, as reflected in the electromyography root mean square value [36]. However, most existing fatigue-related studies have focused on either physical or mental fatigue in certain research fields. Taking physical and mental fatigue as independent variables, different fatigue monitoring indicators were proposed and applied. Furthermore, few attempts have been made to examine the correlation and influence mechanisms between the physical and mental fatigue of farmers in rural tourism, considering the specific characteristics of rural tourism development. According to the above research review, interactions between the two types of fatigue under COVID-19 crisis were explored and proven, and knowledge of the potential mechanisms underlying the development of fatigue was extended. Therefore, we propose the following hypothesis:

**H3.** 
*The interaction between physical and mental fatigue caused by COVID-19 negatively affects farmers’ participation in rural tourism development.*


Then, we conducted a conceptual model to explain how pandemic fatigue affect farmers’ participation in rural tourism development (Figure 1).

## 3. Materials and Methods

### 3.1. Study Sites

The Shanlian Village in Wuxi and the Huashu Village in Nanjing, famous for their rural tourism, offered appropriate locations for conducting the current study. Shanlian Village covers an area of 6.8 square kilometers, located in eastern Wuxi, Jiangsu Province. The village has many unique physical and cultural assets, contributing to its competitiveness as a tourist destination. There are 1740 households, and 90% of the population is involved in rural tourism operations. Its tourism revenue was 13.05 million RMB in 2021. Huashu Village is located in eastern Nanjing and has an area of 9 square kilometers. It is an ancient village with a history of 1500 years, 1140 households, and a total population of 2850. Huashu Village combines several unique agricultural cultures that contribute significantly to its destination competitiveness, making it a very attractive destination for tourists. According to the statistics, 90% of the population in the village recently established tourism-based businesses, and its tourism revenue reached about 10 million RMB in 2021 [37]. Overall, both villages are rich in tourism resources and engaged in farmhouses, gardening, picking, fishing, and bed and breakfasts (B&Bs). In addition, they won the title of “Ecological village” and “the most Beautiful Village” in China (Figure 2). Since 2020, several areas in Shanlian Village have implemented strict epidemic control measures, such as limiting the number of visitors and contact between the residents of confined settings, cancellation, prohibition, and restriction of mass gatherings and smaller meetings, internal or external border closures, and stay-at-home restrictions for all regions. In contrast, Huashu Village was less affected by the epidemic, and there were no confirmed cases. Therefore, the control measures in Huashu Village were moderate, including calls for social distancing and wearing masks. For both Shanlian and Huashu Villages, the natural environment, social conditions, and development of rural tourism are similar; however, the extent of the impact of COVID-19 is different (Table 1). Therefore, a comparison between the two villages is appropriate to test how pandemic fatigue affects farmers’ participation in rural tourism.

### 3.2. Quasi-Experiment Procedures

Experiments are commonly used in tourism research. Experiments assess the causal effect of intervention X (the independent variable) on outcome Y (the dependent variable). Since most tourism studies related perceptions that surface only during an actual tourism context, this study opted for a quasi-experiment to gain a better understanding of the relationship between pandemic fatigue and farmers’ participation in rural tourism. Alasuutari et al. defined quasi-experiments as experiments in which the treatment is intentional or planned [38]. Quasi-experimental designs offer external validity, meaning that the variables can be manipulated in natural settings, which would otherwise be virtually impossible [39]. A quasi-experiment must be designed and executed to generate valid third-order knowledge. According to the extent of exposure to COVID-19, the quasi-experiment in this study involved an experimental group (farmers in Shanlian Village) and a control group (farmers in Huashu Village). The design was based on one independent variable (farmers’ participation in rural tourism) and two dependent variables (physical and mental fatigue). The role of mental and physical fatigue was manipulated with a factorial plan: two perceived mental fatigue levels (high vs. low) and two perceived physical fatigue levels (high vs. low). In the first pair of scenarios, one included, while the other excluded a higher level of mental fatigue. The second pair of scenarios manipulated peer opinions about respondents’ physical fatigue during the COVID-19 crisis. The quasi-experiment may present limitations for the experimental group, which may differ from the control group in characteristics that are correlated with the results being studied, thereby distorting the impact results [40]. Every effort was made to overcome this challenge for all the observed differences.

According to the severity of the outbreak, the respondents from Shanlian Village were regarded as the experimental group, whereas their counterparts from Huashu Village were the control group (See Figure 3). The questionnaire in the field which has been effectively utilized in tourism research was used as a valid method for data collection. Data were collected from the permanent populations of the two villages that participated in the rural tourism business. This effectively avoids the limitations found in previous studies that utilized a convenient student sample in a laboratory setting with some fatigue stimulus [41]. This study was conducted in accordance with the Declaration of Helsinki and the protocol was approved by the Council of Nanjing Forestry University (reference number NJFU-8022311). The quasi-experiment relied on the convenience/snowball sampling method. It was distributed on-site by trained researchers from 12–30 March 2021. A total of 317 valid responses were collected from Huashu Village, and 526 valid responses were collected from Shanlian Village. Since this study adopts self-reported questionnaire to examine farmers’ perception of pandemic fatigue and participation intention, a sampling rule of thumb is the “ten times rule” [42]; it means a sampling threshold for self-reported questionnaire in the order of ten times that of the survey items. Since the questionnaire consisted of 13 items (see Appendix A), we can safely conclude that 317 and 526 are acceptable sample sizes for the two groups examined in this study. Moreover, this study uses Cohen’s *d* to compare the differences between experimental and control group, which is not affected by the sample size and truly reflects the effect size. In both cases, we can safely conclude that the research sample meets the criterion of the minimum sample size and trust the reliability of the results.

As shown in Figure 3, the questionnaires in all scenarios began with demographic questions. The respondents were then invited to imagine their experience during the COVID-19 crisis and report their level of pandemic fatigue in the past three years. Afterwards, respondents were asked to evaluate their intention to participate in rural tourism when they suffered from the lockdown policy enacted by the respective local governments. The collected information will be used to shed light on how pandemic fatigue affects farmers’ participation in rural tourism. The statistical software SPSS 26.0 was adopted in this study to estimate the conceptual model through a series of quantitative analysis such as Chi-square test, *t*-test, and multi-factor analysis of variance.

Table 2 provides an overview of the sample of the 897 respondents concerning gender, age, and educational background in each scenario (survey condition). Of the 526 farmers who engaged in rural tourism business in Shanlian Village, 54.94% were female. The average age of the participants was 38.42 years (standard deviation (SD) = 8.48), ranging from 20 to 70 years. The majority of the participants had a secondary school or higher education (67.30%). Of the group involved in Huashu village’s rural tourism, 56.33% were female. The average age of participants was 36.21 years (SD = 6.52). Most of them had a secondary school or higher education (78.17%). The analysis showed no significant differences in the demographic characteristics between the two villages, making these two sets of data suitable for comparing the effects of different degrees of pandemic fatigue.

### 3.3. Measurements

The questionnaire comprised three main parts (Appendix A). All of the survey items were developed through a review of the available literature, and the actual situation of rural tourism in China. The first part investigated respondents’ demographic characteristics. The second part concerned farmers’ fatigue during COVID-19, and the third part concerned their participation intention. All the items in the second and third parts were measured using a 5-point Likert scale, which was used to examine the respondents’ level of the agreement with various statements appearing in the questionnaire. Although the survey statements were primarily derived from English language literature, we adjusted for this according to the situation of rural tourism in China.

The Pandemic Fatigue Questionnaire was used to assess farmers’ mental and physical fatigue associated with the COVID-19 pandemic [43]. This scale was developed by Labraque containing two dimensions, namely physical fatigue and mental fatigue. It was originally designed to assess pandemic fatigue in the young adult population of the Philippines [44]. Respondents were asked to describe their perception in face of the COVID-19 pandemic, such as “I have been experiencing headaches and body pains” and “I have been experiencing a general sense of emptiness”. This scale has been confirmed and verified by measuring fatigue levels in different populations.

In the third section, the farmers’ willingness to participate in rural tourism development was measured by three statements using a Likert-type scale (1 = strongly disagree, 5 = strongly agree). As a direct expression of willingness to participate, direct participation in the development of rural tourism involves employment and investment in the development of rural tourism. The first captures farmers’ willingness to participate directly in the development of rural tourism [45]. The second part of the questionnaire was focused on supporting the promotion of local rural tourism, which was expressed by recommending others to participate in the development of local rural tourism [46]. In addition, the benefits of promoting rural tourism to the outside world are conducive to broader social investment and participation in rural tourism. Relaying the positive message of rural tourism to others formed the third part, and was related to willingness to participate [47].

Cross-analysis and chi-square tests were employed to determine whether there were differences in the demographic characteristics of physical fatigue, mental fatigue, and willingness to participate in rural tourism development. The results indicated that there was no significant difference in these three constructs according to gender, age, and educational background. Therefore, the interference of demographic characteristics on the different groups of respondents can be eliminated.

## 4. Results

### 4.1. Common Method Variance

Common method variance (CMV) can be a significant concern in self-administered surveys when the same participant responds to all the questionnaire items. It is problematic if a single latent factor accounts for the majority of the explained variance. This study first applied Harman’s single-factor test, as recommended by Podsakoff et al., to account for potential CMV [48]. Harman’s test returned a multi-factor rather than a single-factor solution, and the first factor explained only 18.3% of the total variance. Since the first factor did not capture most of the variance, it was revealed that common method bias was not a severe issue and can thus be disregarded in this study. Due to the limitations of Harman’s single-factor test, a latent variable approach may be more feasible to implement because it does not require a priori identified causes of bias or marker variables. A single unmeasured first-order factor (i.e., common factor) was added to a second confirmation factor analysis (CFA) with all of the measures as indicators. Next, standardized regression weights for all loadings across the two models were compared. Significant differences were also not found that would indicate common method bias.

### 4.2. Validity and Reliability

The test results indicated that both the reliability and validity of the scale were ideal. Table 3 presents the data for Cronbach’s alpha, whose threshold values were similar to those for composite reliability. Internal consistency reliability was assessed using the Joreskog’s composite reliability indicator. The reliability is considered satisfactory to good when the values of mental fatigue, physical fatigue, and participation in rural tourism are between 0.70 and 0.90 [49]. This study also used rho, a value proposed by Dijkstra and Henseler, which is considered an approximately accurate measure of construct reliability. This indicator usually takes a value between Cronbach’s alpha and composite reliability [50]. Table 4 shows that the values of these indicators exceeded 0.7, thereby indicating their reliability. The average variance extracted (AVE) of all the items of each construct was used to measure convergent validity. An AVE is considered acceptable if its value is equal to or greater than 0.50, which suggests that the construct explains at least 50% of the variance in its items [51]. The model constructs obtained AVE values of 0.701 for physical fatigue, 0.814 for mental physical fatigue, and 0.739 for farmer participation, all of which exceeded the recommended thresholds (see Table 3).

As indicated in Table 4, farmers in Shanlian Village experienced more mental fatigue than farmers in Huashu Village (M_experimental group_ = 4.85, M_control group_ = 3.82), whereas the mean values for farmers’ participation in rural tourism development were lower for Shanlian Village than for Huashu Village(M_experimental group_ = 4.17, M_control group_ = 4.83). The farmers’ physical fatigue in Shanlian Village is also higher than their counterparts in Huashu village (M_experimental group_ = 3.85, M_control group_ = 3.68). However, the gap of physical fatigue between the two groups was not as large as mental fatigue. These findings indicate that COVID-19 had a slightly more negative influence on Shanlian Village than on Huashu Village, especially when it comes to mental health. As a result, Huashu Village farmers were more inclined to participate in the process of rural tourism development than Shanlian Village farmers. Thus, the selection of the two locations for the quasi-experiment was valid.

### 4.3. Disturbing Variables

Consistent with the procedure recommended by Armstrong and Overton, F-test were used to examine whether the relationships between pandemic fatigue and farmers’ participation are different in social demographic characteristics [52]. The F-test showed that the sig values are all among [0.288,0.798], higher than the threshold of 0.05. The result revealed no significant differences in mental fatigue, physical fatigue, and farmers’ participation in rural tourism industry among respondents of different gender, age, and education. Therefore, the disturbing effects of these demographic characteristics on the relationship between pandemic fatigue and farmers’ participation can be eliminated.

### 4.4. The Effects of Mental and Physical Fatigue on Farmers’ Participation in Rural Tourism Industry

To confirm whether mental fatigue is negatively related to farmers’ participation in rural tourism, a paired sample *t*-test was conducted. The results presented in Table 5 confirm that farmers’ participation is significantly lower for Shanlian Village (M_experimental group_ = 4.17) than for Huashu Village (M_control group_ = 4.83), *t* = 2.82, *p* = 0.004. This supports hypothesis H1, as Farmers’ mental fatigue has a significant and negative effect on their intention to participate in rural tourism development.

To examine the influence of physical fatigue on farmers’ participation, this study divided the respondents into two groups based on the median value of physical fatigue. Therefore, the respondents from the two villages were divided into two groups with the same number. The mean value of the higher physical fatigue group (M_higher physical fatigue group_ = 4.66) was significantly higher than that of the low physical fatigue group (M_low physical fatigue group_ = 3.52), *t* = 9.58, *p* < 0.001, indicating that the grouping of physical fatigue in this study was valid. The *t*-test results revealed significant differences between the two groups with respect to the effect of physical fatigue on farmers’ participation (M_high physical fatigue group_ = 4.79, M_low physical fatigue group_ = 3.88; *t* = 5.21, *p* < 0.001). This directly confirmed that the more physical fatigue farmers perceived, the less they participated in rural tourism development, thus supporting hypothesis H2.

### 4.5. The Interaction Effects of Physical and Mental Fatigue

A 2 (mental fatigue low or high) × 2 (physical fatigue low or high) MANOVA was adopted to test the interaction effects of physical and mental fatigue on farmers’ participation [53,54]. In Table 6, the results revealed that the main effects of mental fatigue (*F* = 5.63, *p* = 0.001) and physical fatigue (*F* = 4.37, *p* = 0.003) were both significant, further validating hypotheses H1 and H2, suggesting that pandemic fatigue prevents farmers from participating in rural tourism development. Additionally, the interaction effects of physical and mental fatigue were significant (*F* = 11.78, *p* < 0.000).

In order to clearly reveal the interaction effects of mental and physical fatigue on farmers’ participation in rural tourism, the simple effects analysis was adopted [55]. As shown in Table 7, when the degree of physical fatigue was low, the experimental group with farmers with a higher level of mental fatigue (M = 3.87, SD = 1.03) was significantly lower than that of the control group with farmers with low levels of mental fatigue (M = 4.72, SD = 0.70), *t* = 5.21, *p* < 0.001. When the degree of physical fatigue was higher, the experimental group with higher mental fatigue had a low value on their participation (M = 3.51, SD = 1.04), whereas the control group with low mental fatigue had a higher value on farmers’ participation (M = 4.36, SD = 1.26), and the difference between the two groups was significant (*p* < 0.001). Therefore, for farmers with low physical fatigue, the higher their mental fatigue, the less willing they were to participate in rural development. Conversely, for the higher physical fatigue group, farmers with low levels of mental fatigue were still more willing to participate in rural tourism development. Only when both physical and mental fatigue are low, the intention of farmers to participate in rural tourism is the highest. There is a negative relationship between pandemic fatigue and farmer participation.

## 5. Discussion

Since the emergence of the pandemic in late 2019, fatigue has become a prominent issue for the public. This study aimed to assess an integrated view of the combined effects of different kinds of pandemic fatigue, such as physical and mental fatigue, on farmers’ participation in rural tourism development. This quasi-experiment, a methodology used rarely in tourism research found evidence for a direct negative relationship between mental and physical fatigue during the pandemic and farmers’ rural tourism participation (H1 and H2). Furthermore, this study argues that the interaction effects of mental and physical fatigue significantly influence farmers’ participation intention in rural tourism development during the COVID-19 crisis (H3). Overall, our findings support the proposed hypothesized model and address the gap in understanding the mechanism of pandemic fatigue on farmers’ participation in rural tourism development. Considering the relationships among these constructs, administrators should highlight the critical role of farmers’ health and well-being in rural tourism, especially the effects of mental health, which could yield great benefits.

### 5.1. Theoretical Implications

The results of the current study are in line with international studies [56,57,58] that showed individuals who live and work during the COVID-19 pandemic experience moderate to high levels of fatigue. Data from the literature converge on the assumption that fatigue is a complex phenomenon [59]. In the experimental group (farmers in Shanlian Village), residents were more severely affected by the pandemic, and the restrictions to control the disease were stricter. Therefore, their level of physical and mental fatigue since the start of the pandemic was higher than that of the control group (farmers in Huashu Village). These findings support the idea that various lockdown measures, such as social distancing, quarantines, and stay-at-home orders, further worsen people’s fatigue [60]. Additionally, our study found that the mental fatigue scores for the two villages were significantly different, and the level of mental fatigue was higher than that of physical fatigue in both villages. The findings indicate that, compared with the negative impacts of COVID-19 on the body and physiology, such as body weakness, sleep problems, and immunity, farmers’ mental fatigue is a more prominent issue for the public and should not be overlooked in hard-hit villages. Given the adverse consequences of pandemic fatigue, it is essential that institutional measures are implemented to address this issue in rural areas and promote farmers’ overall health and well-being.

The current study revealed that an increased level of pandemic fatigue adversely affected farmers’ participation in rural tourism development. Our results support earlier research which showed that many individuals who experienced dissatisfaction, boredom, and tiredness increased their intention to leave their work because of the threat posed by the pandemic [61,62]. Moreover, evidence from the pre-pandemic period also showed that persistent exposure to physical and mental fatigue might negatively affect individuals’ participation, such as reducing their job engagement, co-creation intention, and investment motivation, and increasing the intent to leave the organization and the profession [63]. If the issue of pandemic fatigue in farmers is not addressed, this may eventually drive farmers away from rural tourism, further worsening staffing problems and negatively affecting the economic and social development of rural areas, leading to low farmer satisfaction and, possibly, more instances of social conflict.

The MANOVA analysis provides additional knowledge concerning the interaction effects of mental and physical fatigue on farmers’ participation in rural tourism development. When the degree of physical fatigue is low, farmers with higher levels of mental fatigue are less willing to participate in rural tourism. Conversely, when the degree of physical fatigue is higher, farmers with low levels of mental fatigue are more willing to participate in this industry. When both physical and mental fatigue are low, farmers’ intention to participate in rural tourism development reaches its highest level. It was expected that the COVID-19 pandemic would not only affect physical health, but also mental health and well-being [64,65]. During the COVID-19 crisis, both mental and physical fatigue are stressors, leading to farmers on the frontline of rural tourism having more fear of the disease, psychological distress, emotional exhaustion, and depression, and reducing their quality of life [66,67]. Furthermore, the results showed that mental fatigue has a more negative effect on farmers’ participation than physical fatigue. This confirmed the findings of previous studies that mental fatigue is regarded as a key factor impacting participation, which is closely linked to low work efficiency, increased risk of error, and even chronic and life-threatening issues. As Hakim et al. stated, physical fatigue may reduce as the pandemic ends, but the negative effects of mental fatigue on farmers’ participation will persist for a long time [68].

### 5.2. Practical Implications

Policymakers and academics have been debating the idea of farmers’ participation in rural tourism development to meet the goals of sustainability for decades, and there is consensus on its value [69,70]. As farmers’ participation is constrained by pandemic fatigue during the COVID-19 crisis, targeted recommendations for fatigue management can be proposed to promote rural tourism. In practice, tourism activities require both physical and mental efforts. To promote farmers’ participation in rural tourism, physical workloads, and intensity during the COVID-19 crisis need to be examined and managed with respect to fatigue management. Compared to physical fatigue, mental fatigue was relatively dominant and qualitative, which is difficult to manage from the perspective of task intensity and workload [71]. Therefore, farmers must pay close attention to their psychological health and well-being to identify potential cognitive tiredness. Furthermore, targeted measures for fatigue management should be proposed in the future based on personal differences.

### 5.3. Limitations of This Study

The current study has some limitations. First, self-report measures were adopted because they allow for the fast collection of data from individuals’ direct responses towards pandemic fatigue. In future research on the interactions between physical and mental fatigue, a comprehensive approach (e.g., a measurement approach utilizing physiological or biomechanical indicators) could be explored for more objective and real-time physical fatigue recognition and measurement. Second, this study only examines the disturbing effects of social demographic characteristics on the relationship between pandemic fatigue and farmers’ participation in rural tourism industry. Future research should consider more disturbing variables such as farmers’ physical and mental health. Thereby, we would be able to capture the mechanism of pandemic fatigue on farmers’ participation intention. Further, this study treated the farmers from each village as a whole while neglected the potential personal differences in practice. A larger group of subjects (particularly specific farmer groups) is recommended for future work to improve the validity of the research outputs. Lastly, this study relied on a convenience sampling to collect data which may not be extrapolated to the whole population. A well-designed systematic cluster sampling based on different population concentrations and characteristics in the future may provide an adequate sample to be representative of the farmers in Chinese villages. Thus, targeted and personalized fatigue management measures can be developed to provide practical benefits in promoting farmers’ participation in rural tourism development.

## 6. Conclusions

This study represents the first step in understanding the constraint mechanism of pandemic fatigue on farmers’ participation in rural tourism development. Identifying the interaction effects of mental and physical fatigue is integral to this process. Two Chinese villages, Shanlian Village in Wuxi and Huashu Village in Nanjing, were chosen as locations for a quasi-experimental survey. Through intra-group comparisons, the negative effects of pandemic fatigue were analyzed according to the conservation of resources theory. The research results also reduce the gap in previous studies by showing how the interaction between physical and mental fatigue affects farmers’ participation. For farmers with low physical fatigue, the higher their mental fatigue, the less willing they were to participate in rural development. Conversely, for the higher physical fatigue group, farmers with low levels of mental fatigue were still more willing to participate in rural tourism development. Overall, the mental and physical fatigue of the farmers at the case sites influenced their willingness to engage in rural tourism during the COVID-19 crisis. This research provides strong support for a multisensory connection between farmers and the pandemic. The results suggest that targeted recommendations and fatigue countermeasures can be proposed to promote farmers’ participation intentions in rural tourism development.

## Figures and Tables

**Figure 1 ijerph-20-00062-f001:**
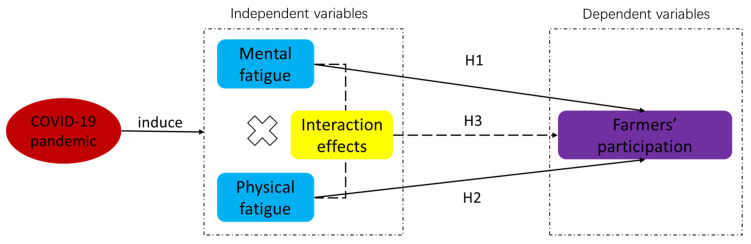
A conceptual model of hypothesized relationships. Note: the red oval represents the stimuli, the blue rectangles represent dependent variables, the grey rectangle represents independent variables, and the yellow rectangles represent the interaction effects of mental and physical fatigue.

**Figure 2 ijerph-20-00062-f002:**
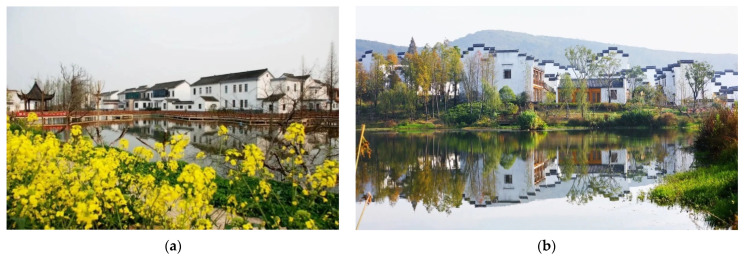
Rural tourism in the study sites. (**a**) B&B in Shanlian Village. (**b**) Agricultural attractions in Huashu Village.

**Figure 3 ijerph-20-00062-f003:**
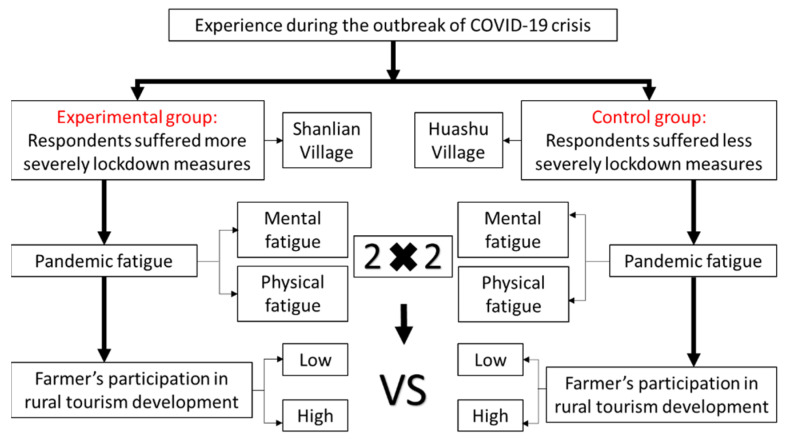
Flowchart of the experimental design.

**Table 1 ijerph-20-00062-t001:** A comparison between Shanlian Village and Huashu Village.

Character	Shanlian Village	Huashu Village
Locations	Jiangsu Province	Jiangsu Province
Population	1740 households	1140 households
Annual tourism income	13.05 million	10 million
Tourism attractions	Farmhouses, gardening, picking, fishing, and B&B	Farmhouses, gardening, picking, fishing, and B&B
Impacts of COVID-19	Three residents were infected by the epidemic, 18 unincorporated villages were locked down in 2020, and all of the 40 unincorporated villages were locked down in 2021	Non-residents were infected by the epidemic, 5 unincorporated villages were locked down in 2020, and 8 unincorporated villages were locked down in 2021

**Table 2 ijerph-20-00062-t002:** Profile of respondents.

Characteristics	Frequency		Percentage (%)	
	Experimental Group (Shanlian Village)	Control Group (Huashu Village)	Experimental Group (Shanlian Village)	Control Group (Huashu Village)
Gender				
Male	237	122	45.06	43.67
Female	289	249	54.94	56.33
Year of birth				
1990–1999	98	74	18.61	19.95
1980–1989	142	106	27.03	28.57
1970–1979	105	71	20.01	19.14
1960–1969	83	43	14.75	11.59
1950–1959	68	39	11.94	10.51
Before 1950	40	38	7.66	10.24
Level of education				
No formal education	22	7	4.18	1.89
Primary school	45	26	8.56	7.01
Secondary school	84	41	15.97	11.05
Diploma and above	354	290	67.30	78.17
Other	21	7	3.99	1.89

**Table 3 ijerph-20-00062-t003:** Individual item reliability and validity.

Construct	Cronbach’s Alpha	Composite Reliability	Rho_A	AVE
Physical fatigue	0.859	0.902	0.863	0.701
Mental fatigue	0.742	0.851	0.822	0.814
Farmers’ participation	0.801	0.887	0.831	0.739

**Table 4 ijerph-20-00062-t004:** Descriptive analysis.

	M (SD)	
	Experimental Group (*n* = 187)	Control Group (*n* = 139)
Physical fatigue	3.85 (1.46)	3.68 (1.60)
Mental fatigue	4.85 (1.02)	3.82 (1.35)
Farmers’ participation	4.17 (0.78)	4.83 (1.11)

**Table 5 ijerph-20-00062-t005:** Test of differences.

	*t*	*df*	*p*	Cohen’s *d*
Physical fatigue	0.31	434	0.125	0.10
Mental fatigue	9.23	434	0.000	2.13
Farmers’ participation	2.82	434	0.004	1.27

**Table 6 ijerph-20-00062-t006:** Multi-factor analysis of variance.

Variable	*df*	*F*	*p*	*η* ^2^	*R* ^2^
Mental fatigue	1	5.63	0.001	0.07	
Physical fatigue	1	4.37	0.003	0.06	0.18
Physical fatigue × mental fatigue	1	11.78	0.000	0.12	
Error	432				

**Table 7 ijerph-20-00062-t007:** Simple effects analysis of farmers’ participation in rural tourism development.

Groups	Low Physical Fatigue Group	Higher Physical Fatigue Group	*p*
Control group (Low mental fatigue)	4.72 (0.70)	4.36 (1.26)	0.000
Experimental group (Higher mental fatigue)	3.87 (1.03)	3.51 (1.04)	0.000

## Data Availability

The original data are provided by all the authors. If there are relevant research needs, the data can be obtained by sending an email to Mengyuan Qiu (qiumengyuan@njfu.edu.cn). Please indicate the purpose of the research and the statement of data confidentiality in the email.

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
