# Peer review of "Does Pandemic Fatigue Prevent Farmers’ Participation in the Rural Tourism Industry: A Comparative Study between Two Chinese Villages"

_ijerph, 2022, doi:10.3390/ijerph20010062_

Round 1

Reviewer 1 Report

The topic is interesting. The topic is important. The authors proved the existence of a cognitive niche. A science experiment is very important. Note: Each experiment should meet the rigorous criteria of a specific test method.
In the introduction, the research tools used should be briefly described.
Research hypotheses 1 and 2 are trivial (obvious). They should be clarified or detailed. It should be noted that this is a pandemic.
Hypotheses should probably not be in the introduction. The right place for hypotheses is the chapter related to research methodology.
It should be justified whether the research sample meets the criterion of the minimum sample size (appropriate calculations). With what probability can you trust the results of the study. The limitations described in point 4.3. they take away the scientific dimension of the article. The reliability of the results is questionable.

The authors write: "The results presented in Table 5 confirm...". There is no Table 5 in the article.
Figure 4 is redundant. These are basic stats. The figure does not show the effects (as the authors write).

Author Response

Dear reviewer,

  We sincerely appreciate your insightful and constructive comments and suggestions on the manuscript. The report attached summarizes our responses to all the comments point-by-point. All the revisions to the manuscript have been marked up using the “Track Changes” function so that changes can be easily viewed by the editors and reviewers. We hope our revisions further enhance your opinion of the piece. If there are any additional concerns or comments, please let us know. We are more than happy to make any further changes that will improve the paper.

Best regards!

Reviewer 2 Report

The article uses a quasi-experimental approach that is novel in the tourism field. It is therefore an original and interesting article on pandemic fatigue applied to the resident population.

From my point of view there are some issues that need clarification:

-It may be interesting to define the main concept, pandemic fatigue. Is it an adaptation of the WHO definition? What are the characteristics of pandemic fatigue?

-Related to the previous point, it would be important to show the items used to identify such mental or physical fatigue (the questionnaire could be attached in the annexed materials).

-Do the authors believe that there may be disturbing variables in the relationship they establish between pandemic fatigue (physical or mental) and farmers' participation in tourism?

-Harman's single factor has some limitations to identify the common methods variance some researchers consider it an exploratory method and not a statistical test. Could you explain the reason for its use?

- The authors have used sociodemographic variables, I think it is interesting to indicate their relationship with mental fatigue in the two groups.

-The issue of convenience sampling offers certain limitations, although the authors themselves recognize them, the study could not be extrapolated to the whole population. Have you thought of implementing a quota and route procedure typical of sociology or another type of probability sampling in the future?

-There are some minor problems such as excessive paraphrasing of some articles, as in the case of Labraque L. J, 2021 in lines 254 to 260, for example.

Author Response

(The authors gave the same response as above.)

Reviewer 3 Report

Tourism is a very broad concept and the problems associated with its organization can be quite surprising, as exemplified in this article. The study presented in the text indicates post-pandemic trends related to the physical and mental fatigue of farmers, which inhibit the development of tourism in rural areas. The study is presented in a very fair way. Its conduct and methodology do not raise any doubts, but the goal is disturbing and requires additional explanation. It would be worth clarifying how the presented results would affect tourism in China. It would be advisable to present a slightly more detailed background related to the study, not taking into account Chinese regulations, but also global trends related to rural tourism. The very structure of the article in terms of research does not raise any doubts. The presented theses are clear. The diagrams and description of the research tools used are clear. The results are presented clearly and comprehensibly. From the formal point of view, it would be advisable to supplement the results with additional recommendations related to the presented research narrative. However, this would require the above-mentioned clarification of the purpose.

Author Response

(The authors gave the same response as above.)

Round 2

Reviewer 1 Report

The authors made changes as recommended. Not all comments have been taken into account. However, the current shape of the article is acceptable.

Every article can always be improved.

Author Response

  We sincerely appreciate the reviewers’ insightful comments and suggestions on the previous version of this manuscript. We are sorry for neglecting some of the reviewer’s comments on Figure 4. The reviewer mentioned in Round 1that Figure 4 is redundant. These are basic stats. The figure does not show the effects (as the authors write). According to the reviewer’s suggestion, we deleted Figure 4 and added more sentences and tables to explain the results of simple effects analysis (See in line 484-507). We hope our work incorporate the reviewer’s concern and improve the reliability of the manuscript.

  In addition, we checked the references, grammar, spelling, punctuation of the manuscript again. We hope our revisions further enhance his/her opinion of the piece. If there are any additional concerns or comments, please let us know. We are more than happy to make any further changes that will improve the paper.
